# Feet Segmentation for Regional Analgesia Monitoring Using Convolutional RFF and Layer-Wise Weighted CAM Interpretability

**Juan Carlos Aguirre-Arango** * , **Andrés Marino Álvarez-Meza** and **German Castellanos-Dominguez**

Signal Processing and Recognition Group, Universidad Nacional de Colombia, Manizales 170003, Colombia; amalvarezme@unal.edu.co (A.M.Á.-M.); cgcastellanosd@unal.edu.co (G.C.-D.)
* Correspondence: jucaguirrear@unal.edu.co

**Abstract:** Regional neuraxial analgesia for pain relief during labor is a universally accepted, safe, and effective procedure involving administering medication into the epidural. Still, an adequate assessment requires continuous patient monitoring after catheter placement. This research introduces a cutting-edge semantic thermal image segmentation method emphasizing superior interpretability for regional neuraxial analgesia monitoring. Namely, we propose a novel Convolutional Random Fourier Features-based approach, termed CRFFg, and custom-designed layer-wise weighted class-activation maps created explicitly for foot segmentation. Our method aims to enhance three well-known semantic segmentation (FCN, UNet, and ResUNet). We have rigorously evaluated our methodology on a challenging dataset of foot thermal images from pregnant women who underwent epidural anesthesia. Its limited size and significant variability distinguish this dataset. Furthermore, our validation results indicate that our proposed methodology not only delivers competitive results in foot segmentation but also significantly improves the explainability of the process.

**Keywords:** infrared thermal segmentation; regional neuraxial analgesia; deep learning; random fourier features; class activation maps





## 1. Introduction

The use of regional neuraxial analgesia for pain relief during labor is widely acknowledged as a safe method [1]. It involves the administration of medication into the epidural or subarachnoid space in the lower back. This procedure blocks pain signals from the uterus and cervix to the brain. This method is considered safe and effective for most women and is associated with lower rates of complications than other forms of pain relief [1,2]. Electrophysiological testing measures nerve fiber reactions to painful stimuli with electromyography, excitatory or inhibitory reflexes, evoked potentials, electroencephalography, and magnetoencephalography [3]. In addition, imaging techniques objectively measure relevant bodily function patterns (such as blood flow, oxygen use, and sugar metabolism) using positron emission tomography (PET), single-photon emission computed tomography (SPECT), and functional magnetic resonance imaging (fMRI) [4].

Nonetheless, imaging techniques can be costly and are generally prohibited in obstetric patients, limiting their use. A cost-effective alternative approach is utilizing thermographic skin images to measure body temperature and predict the distribution and efficacy of epidural anesthesia [5]. This approach is achieved by identifying areas of cold sensation [5]. The use of thermal imaging provides an objective and non-invasive solution to assess warm modifications resulting from blood flow redistribution after catheter placement [6]. However, an adequate assessment requires temperature measurements from the patient's foot soles at various times after catheter placement to accurately characterize early thermal modifications [7,8]. Regarding this, semantic segmentation of feet in infrared thermal images in obstetric environments is challenging due to various factors. Firstly, thermal

images possess inherent characteristics such as low contrast, blurred edges, and uneven intensity distribution, making it difficult to identify objects accurately [9,10]. The second challenge is the high variability of foot position in clinical settings. Additionally, the specialized equipment required for collecting these images and the limited willingness of mothers to participate in research studies resulted in a need for more available samples and the challenge of acquiring annotated data, which is crucial for developing effective segmentation techniques.

Semantic segmentation is crucial in medical image analysis, with deep learning widely used. Fully Convolutional Networks (FCN) [11] is a popular approach that uses Convolutional layers for pixel-wise classification but produces coarse Region of Interest (ROI) and poor boundary definitions for medical images [12]. Likewise, U-Net [13] consists of encoders and decoders that handle objects of varying scales but have difficulty dealing with opaque or unclear goal masks [14]. U-Net++ [15] extends U-Net with nested skip connections for highly accurate segmentation but with increased complexity and overfitting risk. Besides, SegNet [16] is an encoder–decoder architecture that handles objects of different scales but cannot handle fine details. Mask R-CNN [17] extends Faster R-CNN [18] for instance segmentation with high accuracy but requires a large amount of training data and has high computational complexity. On the other hand, PSPNet uses a pyramid pooling module for multi-scale contextual information and increased accuracy but with high computational complexity and a tendency to produce fragmented segmentation maps for small objects [19].

Specifically for semantic segmentation of feet from infrared thermal images, most works were developed in the context of diabetic foot disorders. In [20], the authors combine RGB, infrared, and depth images to perform plantar foot segmentation based on a U-Net architecture together with RANdom SAmple Consensus (RANSAC) [21], which relies too much on depth information. The authors in [22] use a similar approach to integrating thermal and RGB images to be fed into a U-Net model. Their experiments show that RGB images help in more complex cases. In [23], the authors compare multiple models on thermal images, including U-Net, Segnet, FCN, and prior shape active contour-based methodology, proving Segnet outperforms them all. Similarly, in [24], the authors compare multiple infrared thermographic feet segmentation models using transfer learning and removal algorithms based on morphological operations on U-Net, FCN, and Segnet, showing that Segnet outperforms the rest of the models but with high computational cost.

On the other hand, Visual Transformers (VIT) [25] have revolutionized self-attention mechanisms to identify long-range image dependencies. Several recent works have leveraged VIT capabilities to enhance global image representation. For instance, in [26], a U-Net architecture fused with a VIT-based transformer significantly improves model performance. However, this approach requires a pre-trained model and many iterations. Similarly, in [27], a pure U-Net-like transformer is proposed to capture long-range dependencies. Another recent work [28] suggests parallel branches, one based on transformers to capture long-range dependencies and the other on CNN to conserve high resolution. The authors of [29] propose a squeeze-and-expansion transformer that combines local and global information to handle diverse representations effectively. This method has unlimited practical receptive fields, even at high feature resolutions. However, it relies on a large dataset and has higher computational costs than conventional methods. To address the data-hungry nature of transformer-based models, the work in [30] proposes a semi-supervised cross-teaching approach between CNN and Transformers. The most recent work in this field, Meta Segment Anything [31], relies on an extensive natural database (around 1B images) for general segmentation. However, medical and natural images have noticeable differences, including color and blurriness. It is also pertinent to note that accepting ambiguity can incorporate regions that may not be part of the regions of interest. Specifically, while transformers excel at capturing long-range dependencies, they still face challenges in scenarios where data is scarce [32].

Likewise, transfer learning-based strategies in medical image segmentation is a powerful technique that utilizes pre-trained models to enhance performance, minimize data requirements, and optimize computational resources [33]. Nevertheless, choosing an appropriate and representative pre-trained model is crucial to avoid suboptimal results and potential bias [34,35]. Nevertheless, in our study, we aim to assess the effectiveness of our proposal independently, thus excluding the use of transfer learning.

Here, we present a cutting-edge Convolutional Random Fourier Features (CRFFg) technique for foot segmentation in thermal images, leveraging layer-wise weighted class activation maps. Our proposed data-driven method is twofold. First, it integrates Random Fourier Features within a convolutional framework, enabling weight updates through gradient descent. To assess the efficacy of our approach, we benchmark it against three widely-used architectures: U-Net [13], FCN [11], and ResUNet [36]. We enhance these architectures by incorporating CRFFg at the skip connections, bolsters representation, and facilitate the fusion of low-level semantics from the decoder to the encoder. Second, we introduce a layer-wise strategy for quantitatively analyzing Class Activation Maps (CAMs) for semantic segmentation tasks [37]. Our experimental findings showcase the competitive performance of our models and the accurate quantitative assessment of CAMs. The proposed CRFFg method offers a promising solution for foot segmentation in thermal images, tailored explicitly for regional analgesia monitoring. Additionally, layer-wise weighted class activation maps contribute to a more comprehensive understanding of feature representations within neural networks.

The paper is organized as follows: Section 2 describes the materials and methods used in the study. Sections 3 and 4 present the experimental setup and results, respectively, followed by Section 5, which provides the concluding remarks.

## 2. Material and Methods

### 2.1. Deep Learning for Semantic Segmentation

Provided an image set, $\{ \boldsymbol{I}_n \in \mathbb{R}^{H \times \tilde{W} \times C} : n \in N \}$, we will call a label mask the corresponding matrix $\boldsymbol{M}_n$ that encodes the membership of each $n$-th image pixel to a particular class, where $H$ is height, $\tilde{W}$ is width, and $C$ holds the color channels of the image set. For simplicity, $C = 1$ is assumed. As regards the semantic segmentation task under consideration, each mask is binary, $\boldsymbol{M} \in \{0, 1\}^{H \times \tilde{W}}$, representing either the background or the foreground.

An estimate for matrix mask $\hat{\boldsymbol{M}} \in [0, 1]^{H \times \tilde{W}}$ can be obtained through deep learning models for semantic segmentation, stacking convolutional layers as follows:

$$\hat{\boldsymbol{M}} = (\varphi_L \circ \cdots \circ \varphi_L)(\boldsymbol{I}) \tag{1}$$

where $\varphi_l : \mathbb{R}^{H_{l-1} \times \tilde{W}_{l-1} \times D_{l-1}} \to \mathbb{R}^{H_l \times \tilde{W}_l \times D_l}$ denotes a function composition for the $l$-th layer ($l \in L$), which comprises learnable parameters represented by $\boldsymbol{W}_l \in \mathbb{R}^{\tilde{k}_l \times \tilde{k}_l \times D_{l-1} \times D_l}$ and $\boldsymbol{b}_l \in \mathbb{R}^{D_l}$ ($\tilde{k}_l$ holds the $l$-th convolutional kernel size). Of note, the feature map $\boldsymbol{F}_l = \varphi_l(\boldsymbol{F}_{l-1}) = \varsigma_l(\boldsymbol{W}_l \otimes \boldsymbol{F}_{l-1} + \boldsymbol{b}_l) \in \mathbb{R}^{H_l \times \tilde{W}_l \times D_l}$, is comprised of $D_l$ distinct features extracted, $\varsigma_l(\cdot)$ is a nonlinear activation function, and $\otimes$ stands for image-based convolution. Essentially, the function composition in Equation (1) transforms the input feature map from the previous layer, $(l-1)$, into the output feature map for the current layer, $l$, by employing the learnable parameters $\boldsymbol{W}_l$ and $\boldsymbol{b}_l$. The resulting $\boldsymbol{F}_l$ captures the salient information within the $l$-th network layer.

The parameter set $\Theta = \{\boldsymbol{W}_l, \boldsymbol{b}_l : l \in L\}$ is estimated within the following optimizing framework [38]:

$$\Theta^* = \arg \min_{\Theta} \mathbb{E}\{\mathcal{L}\{\boldsymbol{M}_n, \hat{\boldsymbol{M}}_n | \Theta\} : \forall n \in N\}, \tag{2}$$

where $\mathcal{L} : \{0, 1\}^{H \times \tilde{W}} \times [0, 1]^{H \times \tilde{W}} \to \mathbb{R}$ in Equation (2) is a given loss function and notation $\mathbb{E}\{\cdot\}$ stands for the expectation operator.

### 2.2. Convolutional Random Fourier Features Gradient—CRFFg

Random Fourier Features establish a finite-dimensional, explicit mapping that approximates shift-invariant kernels $k(\cdot)$ as described in Rahimi et al. (2009) [39]. This explicit mapping, denoted by $z : \mathbb{R}^{\tilde{Q}} \to \mathbb{R}^{Q}$, serves to transform the input space into a finite-dimensional space $\mathcal{H} \subset \mathbb{R}^{Q}$, where the inner product can be obtained as:

$$k(\boldsymbol{x} - \boldsymbol{x}') = \langle \phi(\boldsymbol{x}), \phi(\boldsymbol{x}') \rangle_{\mathcal{H}} \approx z(\boldsymbol{x})^{\top} z(\boldsymbol{x}'). \tag{3}$$

The mapping $\boldsymbol{z}$ in Equation (3) is defined through Bochner's theorem [40]:

$$k(\boldsymbol{x} - \boldsymbol{x}') = \int_{\mathbb{R}^{\tilde{Q}}} p(\boldsymbol{\omega}) \exp(i\boldsymbol{\omega}^{\top}(\boldsymbol{x} - \boldsymbol{x}')) d\boldsymbol{\omega} = \mathbb{E}_{\boldsymbol{\omega}}\big\{ \exp(i\boldsymbol{\omega}^{\top}(\boldsymbol{x} - \boldsymbol{x}'))\big\}, \tag{4}$$

where $\boldsymbol{x}, \boldsymbol{x}' \in \mathbb{R}^{\tilde{Q}}$, $p(\boldsymbol{w})$ is the probability density function of $\boldsymbol{w} \in \mathbb{R}^{\tilde{Q}}$ that defines the type of kernel. Specifically, the Gaussian kernel, favored for its universal approximating properties and mathematical tractability [41], is achieved from Equation (4) by setting $p(\boldsymbol{w}) = \mathcal{N}(0, \sigma^2 \hat{\boldsymbol{I}})$; $\sigma \in \mathbb{R}^{+}$ is a length-scale and $\hat{\boldsymbol{I}}$ is an identity matrix of proper size.

As both the kernel and the probability are real values, the imaginary component can be disregarded by employing the Euler equation. This leads to the use of a cosine function rather than an exponential, ensuring the following relationship:

$$z(\boldsymbol{x}) = \sqrt{\frac{2}{Q}} \big[ \cos(\boldsymbol{\omega}_1^{\top}\boldsymbol{x} + b_1), \dots, \cos(\boldsymbol{\omega}_Q^{\top}\boldsymbol{x} + b_Q) \big]^{\top}, \tag{5}$$

where $\boldsymbol{\omega}_q \in \mathbb{R}^{\tilde{Q}}$, $b_q \in \mathbb{R}$, and $q \in Q$.

We aim to extend the kernel-based mapping depicted in Equation (5) for application to spatial data, such as images, by utilizing the power of convolutional operations. These operations have garnered significant attention for their efficacy in processing grid data [42]. Convolutional operations exhibit two crucial properties—translation equivariance and locality—that render them particularly suitable for handling spatial data [42]. In order to integrate these properties into the Random Fourier Features framework, we adapt the $z$ mapping to operate within local regions of the grid input space. This results in the computation of the feature map $\mathbf{F}_l \in \mathbb{R}^{H_l \times \tilde{W}_l \times Q_l}$, where the mapping is defined as $z : \mathbb{R}^{H_{l-1} \times \tilde{W}_{l-1} \times D_{l-1}} \to \mathbb{R}^{H_l \times \tilde{W}_l \times Q_l}$, yielding:

$$\mathbf{F}_l = z(\mathbf{F}_{l-1}) = \cos\left( \frac{\boldsymbol{W}_l}{\Delta_l} \otimes \mathbf{F}_{l-1} + \boldsymbol{b}_l \right), \tag{6}$$

where $\Delta_l \in \mathbb{R}^{+}$ is a scale parameter. The parameters $\boldsymbol{W}_l \in \mathbb{R}^{\tilde{k}_l \times \tilde{k}_l \times D_{l-1} \times Q_l}$ and $\boldsymbol{b}_l \in \mathbb{R}^{Q_l}$ are initialized as in Equations (4) and (5), and updated through gradient descent under a back-propagation-based optimization of Equation (2) [38]. Consequently, we refer to the layers in Equation (6) as Convolutional Random Fourier Features Gradient (CRFFg).

The conceptual depiction of the proposed CRFFg layer is shown in Figure 1. Using this approach, we aim to integrate the advantageous attributes of kernel methods into a deep learning-based feature representation enhancement. In addition, using convolutions for local and equivariant representation of spatial data provides a robust and efficient strategy for image processing.

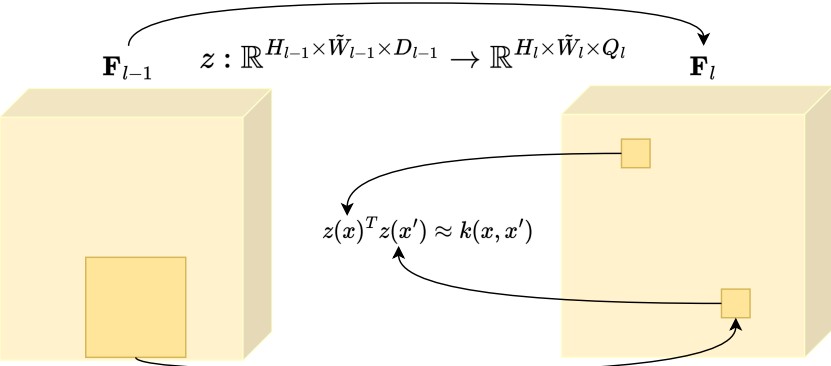

**Figure 1.** The Convolutional Random Fourier Features Gradient (CRFFg) mapping, grounded in kernel methods, is employed for image-based data examination within deep learning frameworks.

### 2.3. Layer-Wise Weighted Class Activation Maps for Semantic Segmentation

Class Activation Maps (CAMs) are a powerful tool to enhance the interpretability of outcomes derived from deep learning models. They achieve this by emphasizing the critical image regions in determining the model's predicted output. To evaluate the contribution of these regions to a specific class $r \in \{0, 1\}$, a linear combination of feature maps from a designated convolutional neural network layer $l$ can be employed [37]. Here, given an input image $I$ and a target class $r$, the salient input spatial information coded by the $l$-th layer into a trained deep learning semantic segmentation model with parameter set $\Theta^*$, as in Equation (2), is gathered through the Layer-CAM algorithm, yielding [43]:

$$S_l^r = (\Lambda \circ ReLU) \left( \sum_{d \in D_l} \boldsymbol{\alpha}_l^{rd} \odot \boldsymbol{F}_l^{rd} \right) \tag{7}$$

where $S_l^r \in \mathbb{R}^{H \times \tilde{W}}$ holds the Layer-CAM for class $r$ at layer $l$, $\Lambda : \mathbb{R}^{H_l \times \tilde{W}_l} \to \mathbb{R}^{H \times \tilde{W}}$ is the up-sampling operator, $ReLU(x) = \max(0, x)$ is the Rectified Linear activation function, and $\odot$ stands for Hadamard product. Besides, $\boldsymbol{F}_l^{rd} \in \mathbf{R}^{H_l \times \tilde{W}_l}$ collects the $d$-th feature map and $\boldsymbol{\alpha}_l^{rd} \in \mathbf{R}^{H_l \times \tilde{W}_l}$ is a weighting matrix holding elements:

$$\alpha_l^{rd}[i,j] = ReLU \left( \partial y^r / \partial F_l^{rd}[i,j] \right), \tag{8}$$

with $\alpha_l^{rd}[i,j] \in \boldsymbol{\alpha}_l^{rd}$ and $F_l^{rd}[i,j] \in \boldsymbol{F}_l^{rd}$. $y^r$ is the score for class $r$ that is computed using the approach in [44] adopted for the semantic segmentation tasks, as follows:

$$y^r = \mathbb{E} \left\{ \tilde{F}_L[i,j] : \forall i,j | M[i,j] = r \right\} \tag{9}$$

where $\tilde{F}_L[i,j] \in \tilde{F}_L$ holds the feature map elements for layer $L$ in Equation (1) fixing a linear activation function.

As previously mentioned, the use of CAM-based representations enhances the explainability of deep learning models for segmentation tasks. To evaluate the interpretability of CAMs for a given model, we propose the following semantic segmentation measures, where higher scores indicate better interpretability:

- CAM-based Cumulative Relevance ($\rho_r$): It involves computing the cumulative contribution from each CAM representation to detect class $r$ within the segmented region of interest. This can be expressed as follows:

$$\rho_r = \mathbb{E}_l \left\{ \mathbb{E}_n \left\{ \frac{\mathbf{1}^\top (\tilde{M}_n^r \odot S_{nl}^r) \mathbf{1}}{\mathbf{1}^\top S_{nl}^r \mathbf{1}} : \forall n \in N \right\} \forall l \in L \right\}, \quad \rho_r \in [0, 1], \tag{10}$$

where $\tilde{M}_n^r \in \{0,1\}^{H \times \tilde{W}}$ collects a binary mask that identifies the pixel locations associated with the class $r$, and $S_{nl}^r$ holds the Layer-CAM for image $n$ with respect to layer $l$ (see Equation (7)).

- Mask-based Cumulative Relevance ($\varrho_r$): It assesses the relevance averaged across the class pixel set related to the target mask of interest. Then, each class-based cumulative relevance is computed as follows:

$$\varrho_r = \mathbb{E}_l \left\{ \mathbb{E}_n \left\{ \frac{\mathbf{1}^\top (\tilde{M}_n^r \odot S_{nl}^r) \mathbf{1}}{\mathbf{1}^\top \tilde{M}_n^r \mathbf{1}} : \forall n \in N \right\} \forall l \in L \right\}, \varrho_r \in \mathbb{R}^+. \tag{11}$$

The normalized Mask-based Cumulative Relevance can be computed as:

$$\rho_r' = \frac{\rho_r'}{\max\limits_{r' \in \{0,1\}} \rho_{r'}}, \quad \rho_r' \in [0,1]. \tag{12}$$

- CAM-Dice ($D'$): A version of the Dice measure that quantifies mask thickness and how the extracted CAM is densely filled:

$$D_r' = \mathbb{E}_l \left\{ \mathbb{E}_n \left\{ 2 \frac{\mathbf{1}^\top (\tilde{M}_n^r \odot S_{nl}^r) \mathbf{1}}{\mathbf{1}^\top \tilde{M}_n^r \mathbf{1} + \mathbf{1}^\top S_{nl}^r \mathbf{1}} : \forall n \in N \right\} : \forall l \in L \right\}, \quad D_r' \in [0,1]. \tag{13}$$

The proposed measures enable the weighting of each layer's contribution to a given class across the model by adjusting the normalization term related to the target mask, the estimated CAM, or both pixel-based salient activations. Figure 2 depicts a graphical representation of the proposed measures. The green circle represents the CAM generated for a specific region, as indicated by the white circle. These measures are designed to capture the relationship between the CAMs and the regions of interest. Furthermore, Figure 3 presents some exemplary scenarios. For instance, the $\rho$ measure is associated with the proportion of the CAM inside the region of interest. On the other hand, $\rho$ is based on the proportion of CAMs that, on average, belong to each pixel of the region of interest while maintaining the relationship between the classes (in this case, green for the foreground and red for the background). Additionally, $D_r'$ follows a similar concept as the Dice coefficient used in segmentation, assessing the homogeneity of the intersection of the regions. In this case, we want to determine if the CAM is uniformly distributed.

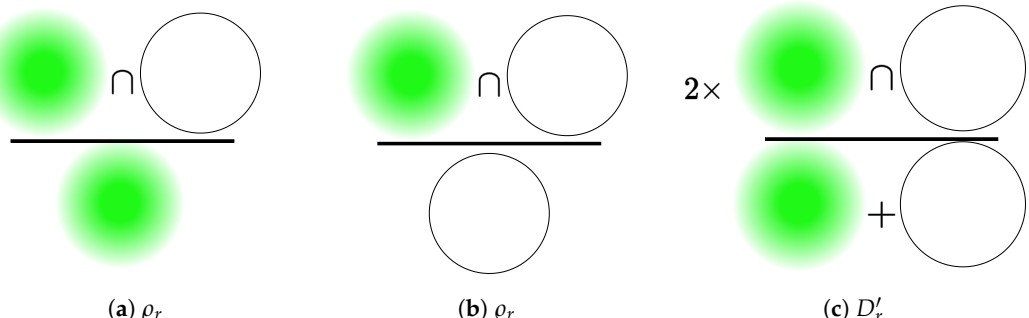

(a) $\rho_r$　　　　　　　　　　　(b) $\varrho_r$　　　　　　　　　　　(c) $D_r'$

**Figure 2.** Graphic depiction of the proposed relevance measures for Layer-Wise Class Activation Maps used in semantic segmentation tasks.

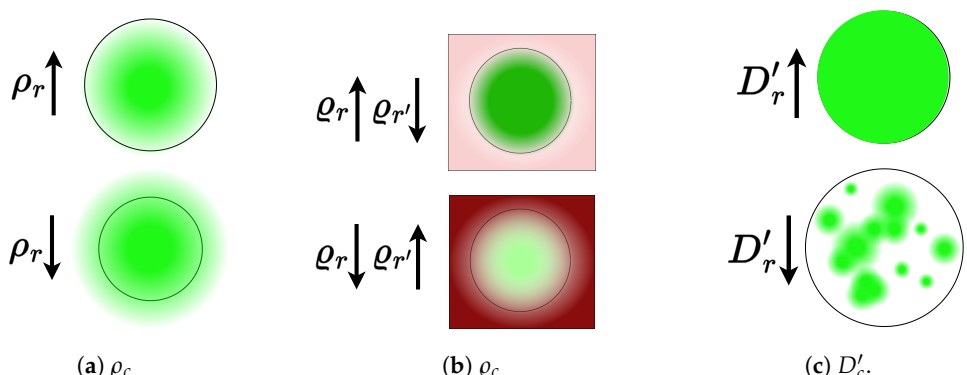

(**a**) $\rho_c$          (**b**) $\varrho_c$          (**c**) $D'_c$.

**Figure 3.** Illustrative scenarios regarding our novel Layer-Wise Class Activation Maps for semantic segmentation.

*2.4. Feet Segmentation Pipeline from Thermal Images*

In a nutshell, the proposed methodology is evaluated using the pipeline shown in Figure 4, including the following testing stages:

(i)    Foot Infrared Thermal Data Acquisition and Preprocessing.
(ii)   Architecture Set-Up of tested Deep models for foot segmentation. Three DL architectures are contrasted using our CRFFg: U-Net, Fully Convolutional Network (FCN), and ResUNet.
(iii)  Assessment of semantic segmentation accuracy. In this study, we examine how data augmentation affects the performance of tested deep learning algorithms.
(iv)   Relevance-maps extraction from our Layer-Wise weighted CAMs to provide interpretability.

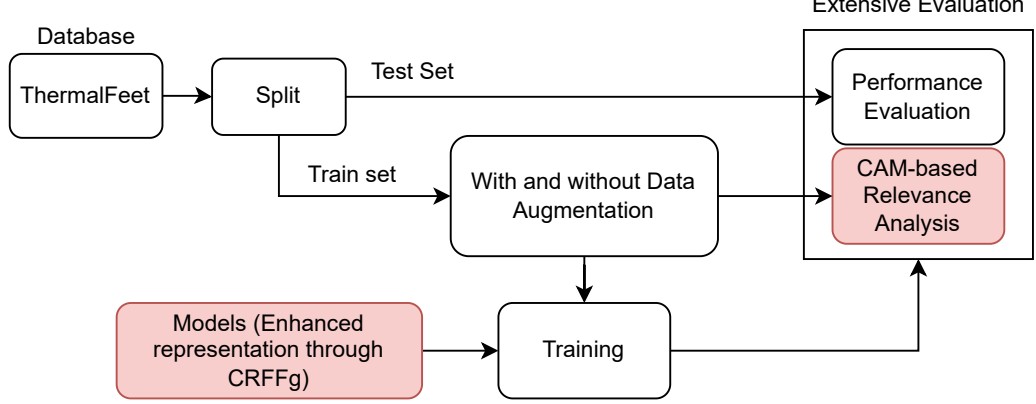

**Figure 4.** Foot segmentation from thermal images using our CRFFg-based deep learning enhancement holding layer-wise weighted CAM interpretability.

**3. Experimental Set-Up**

The proposed deep learning model for semantic segmentation enhances foot thermal images' interpretability, achieving competitive segmentation performance. To this end, we evaluate the impact of incorporating a convolutional representation of CRFFg and layer-wise weighted CAM into three well-known deep-learning architectures.

*3.1. Protocol for Infrared Thermal Data Acquisition: ThermalFeet Dataset*

The protocol for data acquisition was designed by the physician staff at "SES Hospital Universitario de Caldas" to standardize the data collection of infrared thermal images acquired from pregnant women who underwent epidural anesthesia during labor. This protocol is in accordance with the occupational risks associated with assisting local anesthetics via epidural neuraxial as specified by the hospital's administration, following previously implemented protocols [8,45–48].

Patient monitoring includes the necessary equipment for taking vital signs and a metal stretcher with foam cushion and plastic exterior covered only with a white sheet. The continuous monitoring device is placed 1.5 m from the stretcher in the same room, as shown in Figure 5. Before the epidural procedure, anesthesiologists assess each patient clinically and provide written and verbal information about the trial before obtaining her written consent. The patient's body temperature, heart rate, oxygen saturation, and non-invasive blood pressure are monitored every five minutes. Skin temperature values are recorded during the procedure. Sensitivity responses are evaluated using superficial touch and cold tests with cotton wool soaked in water applied to the previously determined dermatomes. The temperature test records the verbal response as Yes or No for superficial touch and Cold or No Cold.

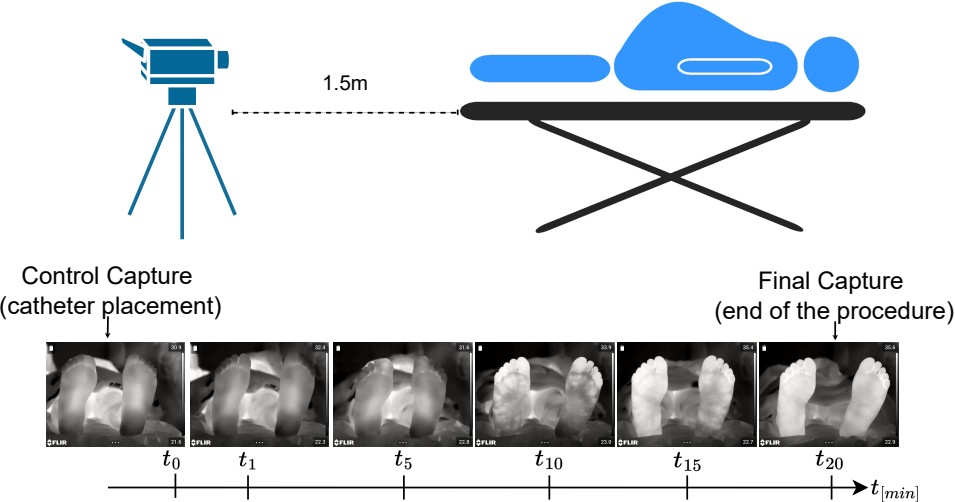

**Figure 5.** Regional analgesia monitoring protocol using local anesthetics via epidural neuraxial and thermal images.

The protocol timeline for acquiring infrared thermal images is as follows: Initially, the woman is asked to be in a supine position before the first thermal image (T0) is captured once the first dose of the analgesic mixture is administered. A single thermal picture is taken at the placement of the operated catheter (0.45 mm; Perifix, Braun®, Kronberg, Germany) positioned within the space selected for injecting epidural anesthesia in the cervical region (at L2 to L3 or L3 to L4), measuring a few millimeters.

Within the next 25 min, one thermographic recording of the lower extremity is taken every five minutes (T1–T5). The catheter remains in the epidural space taped to the skin so that one image is captured every five minutes until six pictures have been collected. Though the clinical protocol demands images of both feet taken in a fixed corporal position, this condition is barely achievable due to the difficulty of labor procedures and contractions.

The data was collected under two different hardware specifications: (i) A set of 196 images captured from 22 pregnant women during labour using a FLIR A320 infrared camera with a resolution of $640 \times 480$ and a spectral range within 7.5 to 13 μm. (ii) A set of 128 images with improved sensitivity and flexibility taken using a FLIR E95 thermal camera, having a resolution of $640 \times 480$ and spectral range within 7.5 to 14 μm. In this study, 166 thermal images are selected from both sets as fulfilling the quality criteria of validation, as detailed in [24]. An anesthesiologist manually segmented the region of interest. The dataset is publicly available at https://gcpds-image-segmentation.readthedocs.io/en/latest/notebooks/02-datasets.html (accessed on 5 April 2023).

### 3.2. Set-Up of Compared Deep Learning Architectures

The following deep learning architectures are contrasted and enhanced using our CRFFg approach:

- Fully Convolutional Network (FCN) [11]: This architecture is based on the VGG (Very Deep Convolutional Network) [49] model to recognize large-scale images. By using only convolutional layers, FCN models can deliver a segmentation map with pixel-level accuracy while reducing the computational burden.
- U-Net [13]: This architecture unfolds into two parts: The encoder consists of convolutional layers to reduce the spatial image dimensions. The decoder holds layers to upsample the encoded features back to the original image size.
- ResUNet [36]: This model extends the U-Net architecture by incorporating residual connections to improve performance. Deep learning training is improved by residual connections, which allow gradients to flow directly through the network.

Figure 6 presents the mentioned architectures, illustrating their unique layers, blocks, and the dimensions and filters associated. Different colors represent the different blocks or layers, and the spatial dimension of each level is also indicated. We estimate the effectiveness of incorporating the CRFFg layer for comparison purposes in FCN, U-Net, and ResUNet architectures. However, each evaluated CRFFg layer arrangement differs from another in the semantic segmentation features that feed the decoder, as detailed in [50–52]. Then, the CRFFg layer is placed at skip connections to enhance the feature fusion between encoders and decoders.

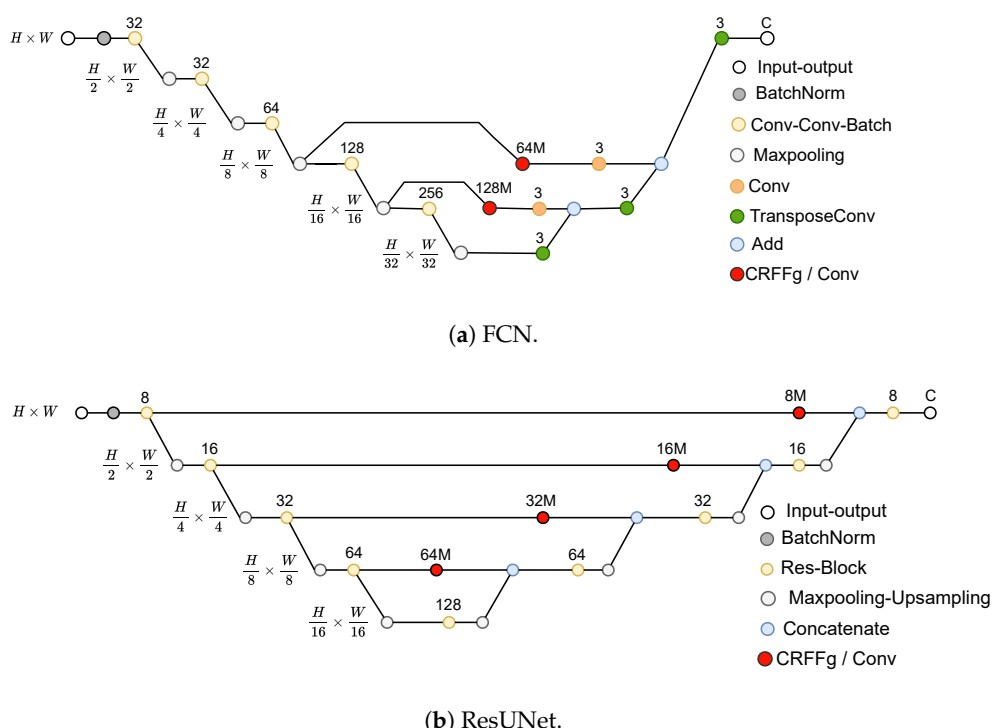

(**a**) FCN.

(**b**) ResUNet.

**Figure 6.** *Cont.*

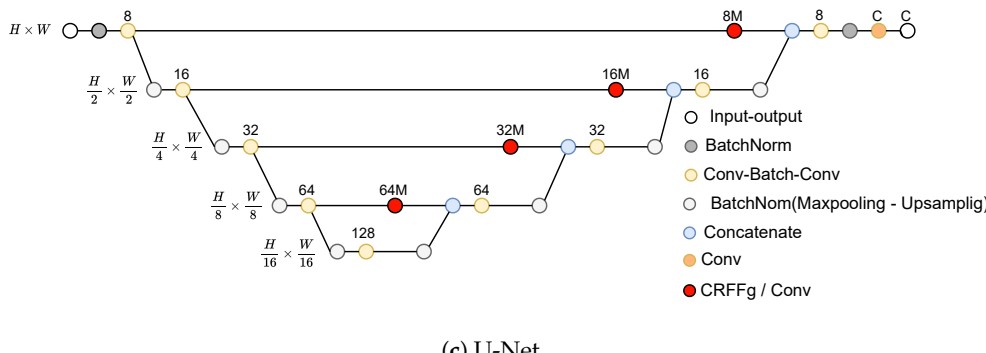

(**c**) U-Net.

**Figure 6.** Tested semantic segmentation architectures. Our CRFFg approach aims to enhance the data representation (see red dots).

To evaluate the performance difference with the proposed CRFFg-layer strategy, we utilize a standard convolutional layer featuring an equal number of filters and a ReLU activation function at the same position within the architecture. In particular, we analyze the influence of the CRFFg layer dimension on segmentation performance, testing two multiplication values (one and three). Besides, to study the impact of CRFFg, we set the hyperparameters of all models variation from FCN, U-Net, and ResUNet architectures the same. The number of epochs is 200, and the batch size is 16. Additionally, the scale value of the CRFFg, $\Delta$, is set as described in the standard RFF's Tensorflow implementation for simplicity. Regarding the weights, they are trained using gradient descent with backpropagation. The selected optimizer is Adam due to its faster convergence, adaptive learning rate, reduced sensitivity to hyperparameters, and combining benefits of convex optimization [53]. The learning rate is initialized as $1e-3$, and a dice-based loss is employed in Equation (2), as follows:

$$\mathcal{L}_{Dice}(\boldsymbol{M}_n, \hat{\boldsymbol{M}}_n) = 2 \frac{\mathbf{1}^\top (\boldsymbol{M}_n \odot \hat{\boldsymbol{M}}_n)\mathbf{1} + \epsilon}{\mathbf{1}^\top \boldsymbol{M}_n \mathbf{1} + \mathbf{1}^\top \hat{\boldsymbol{M}}_n \mathbf{1} + \epsilon}, \tag{14}$$

where $\epsilon = 1$ avoids numerical instability. All experiments are carried out in Python 3.8, with the Tensorflow 2.4.1 API, on a Google Colaboratory environment (code repository: https://github.com/aguirrejuan/Foot-segmentation-CRFFg, accessed on 25 April 2023).

### 3.3. Training Details and Quantitative Assessment

With the aim to prevent overfitting and improve the generalization of trained models, the data augmentation procedure is performed on each image with horizontal flip enabled since feet are mostly symmetrical on the horizontal axis, specifically left-right and right-left on each foot. Hence, vertical overturn is disabled to prevent unrealistic upside-down foot representations. In the augmentation procedure, the images are rotated seven times within a range of $-15$ to $15$ degrees, translated by 10% right to left, and zoomed in and out by 15%, as described in [20].

Moreover, the following metrics are used to measure segmentation performance [54]:

$$D = \frac{2|\boldsymbol{M} \cap \hat{\boldsymbol{M}}|}{|\boldsymbol{M}| + |\hat{\boldsymbol{M}}|} = \frac{2T_P}{2T_P + F_P + F_N} \tag{15a}$$

$$J = \frac{|\boldsymbol{M} \cap \hat{\boldsymbol{M}}|}{|\boldsymbol{M} \cup \hat{\boldsymbol{M}}|} = \frac{T_P}{F_N + F_P + T_P} \tag{15b}$$

$$S_e = \frac{T_P}{T_P + F_N} \tag{15c}$$

$$S_p = \frac{T_N}{T_N + F_P} \tag{15d}$$

where $T_P$, $F_N$, and $F_P$ represent the true positive, false negative, and false positive predictions, respectively, for comparing the actual and estimated label masks $M_n$ and $\hat{M}_n$ for a given input image $I_n$. In addition, the introduced layer-wise, weighted CAM-based interpretability measures are computed for CAM-Dice, CAM-based Cumulative Relevance, and Mask-based Cumulative Relevance (see Equations (12) and (13)).

As for the validation strategy, we selected the hold-out cross-validation strategy with the following partitions: 80% of the samples for training, 10% for validation, and 10% for testing.

## 4. Results and Discussion

### 4.1. Visual Inspection Results

Figure 7 shows results obtained from thermalFeet database without data augmentation, where each row represents a different architecture: FCN in the first row, U-Net in the second row, and ResUNet in the third row. As expected, the performance of the models under a small-size dataset is poor. The regions of faster change in temperature, which characterize the dataset, are where the models struggle more. At first glance, we observe that the FCN architecture is the one that struggles the most, having high false positives regions in regions that exhibit low-high temperatures.

Figure 8 shows results obtained incorporating data augmentation. The positive impact of the data augmentation on the resulting segmentation of all the models is visible. Moreover, FCN architectures produce smoother borders and fewer false positives than other architectures. This can be explained due to the high receptive field that possesses the FCN architecture, allowing it to capture complex and heterogeneous regions (the variability of the temperatures) that compose the feet.

Notably, when comparing FCN models with a multiplication factor of 1 (M1), the model with our CRFFg (blue) generally outperforms in terms of pixel membership prediction (sensitivity). However, this trend only holds when the multiplication factor is increased to 3 (M3), probably because the large model is a propensity to overfit, making the prediction less confident in new data points. On the other hand, U-Net models blunder with regions that exhibit fast temperature changes. The same characteristic the FCN possesses can explain this, but the U-Net does not have a high receptive field that allows it to characterize high heterogeneous feet. As a result, among the U-Net approaches, U-Net CRFFg S-M1 performs satisfactorily with low false positives and high false negatives. At the same time, its direct competitor, U-Net S-M1, shows the opposite trend. Similarly, using CRFFg in the other U-Net alternatives reduces the number of false positives. Finally, the ResUNet architecture has the same behavior as the U-Net but with smoother borders, which can be explained due to the multiple stack layers at the ResBlock, which increase multiple steps of representation, allowing to capture of helpful representation. The ResUNet S-M1 works better on average; adding layers at the skip connections appears to reduce performance, creating false positives and false negatives. The latter can be explained due to the small size of the dataset. Specifically, using CRFFg with ResUNet does not result in noteworthy improvements.

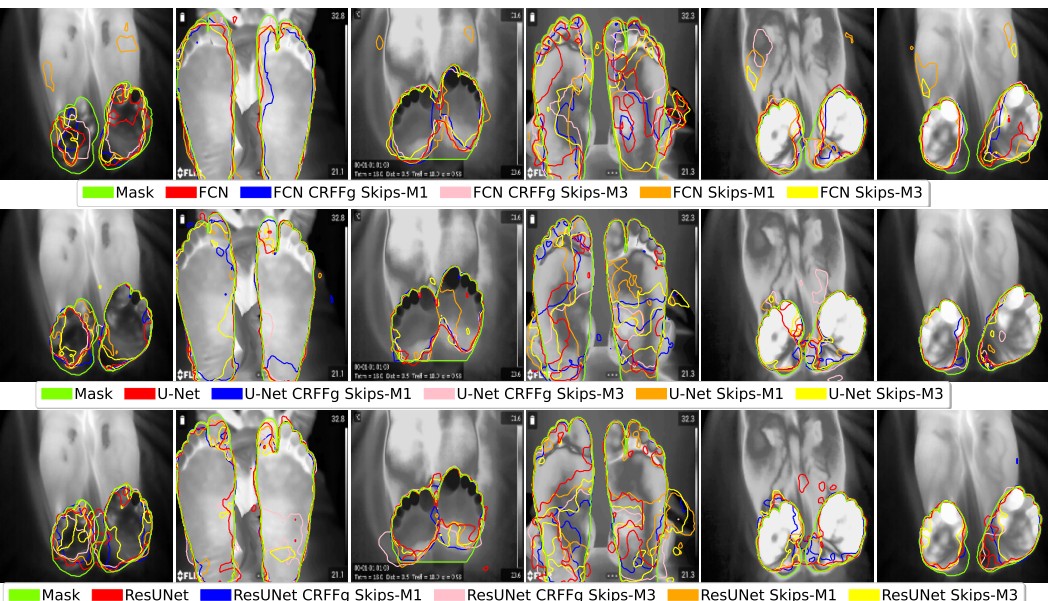

**Figure 7.** Visual inspection of the results on thermalFeet database without data augmentation. Our CRFFg-based enhancements are also presented. The first row shows the results for the FCN architecture, the second row for U-Net, and the third row for ResUNet. A unique color differentiates each model within an architecture. M1 and M3 represent CRFFg's dimension as a multiplication factor of the enhanced layer's size.

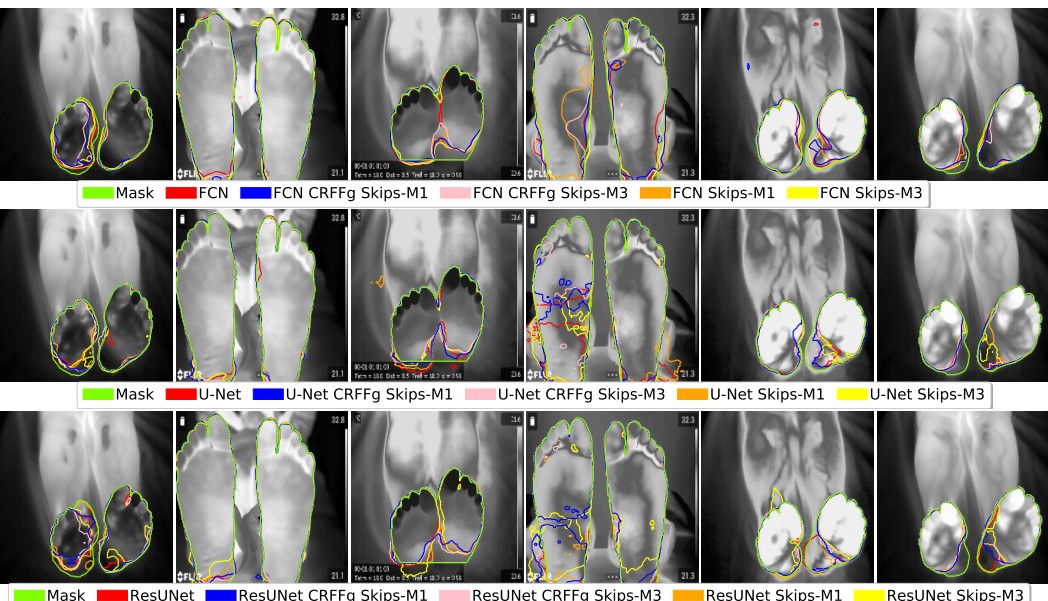

**Figure 8.** Visual inspection of the results on thermalFeet database with data augmentation. Our CRFFg-based enhancements are also presented. The first row shows the results for the FCN architecture, the second row for U-Net, and the third row for ResUNet. A unique color differentiates each model within an architecture. M1 and M3 represent CRFFg's dimension as a multiplication factor of the enhanced layer's size.

### 4.2. Method Comparison Results of Semantic Segmentation Performance

Figure 9 illustrates the learning curves, e.g., training loss vs. epochs, of the compared models. Upon visual inspection, notable differences between the curves with and without data augmentation can be observed. When data augmentation is not applied, the algorithms exhibit higher validation loss in the initial 40 epochs. Regardless, they subsequently demonstrate a downward trend in validation loss. It is essential to mention that the

learning curves exhibit increased noise, likely due to the limited size of the dataset. The limited dataset challenges the models to capture generalized features early in training. Moreover, in the validation partition without data augmentation, some models display a phenomenon known as double descent [55], where layers at different locations in the networks may learn at different rates [56]. In contrast, the training and validation losses consistently decrease in the data augmentation scenario, albeit with minor noise in the validation partition.

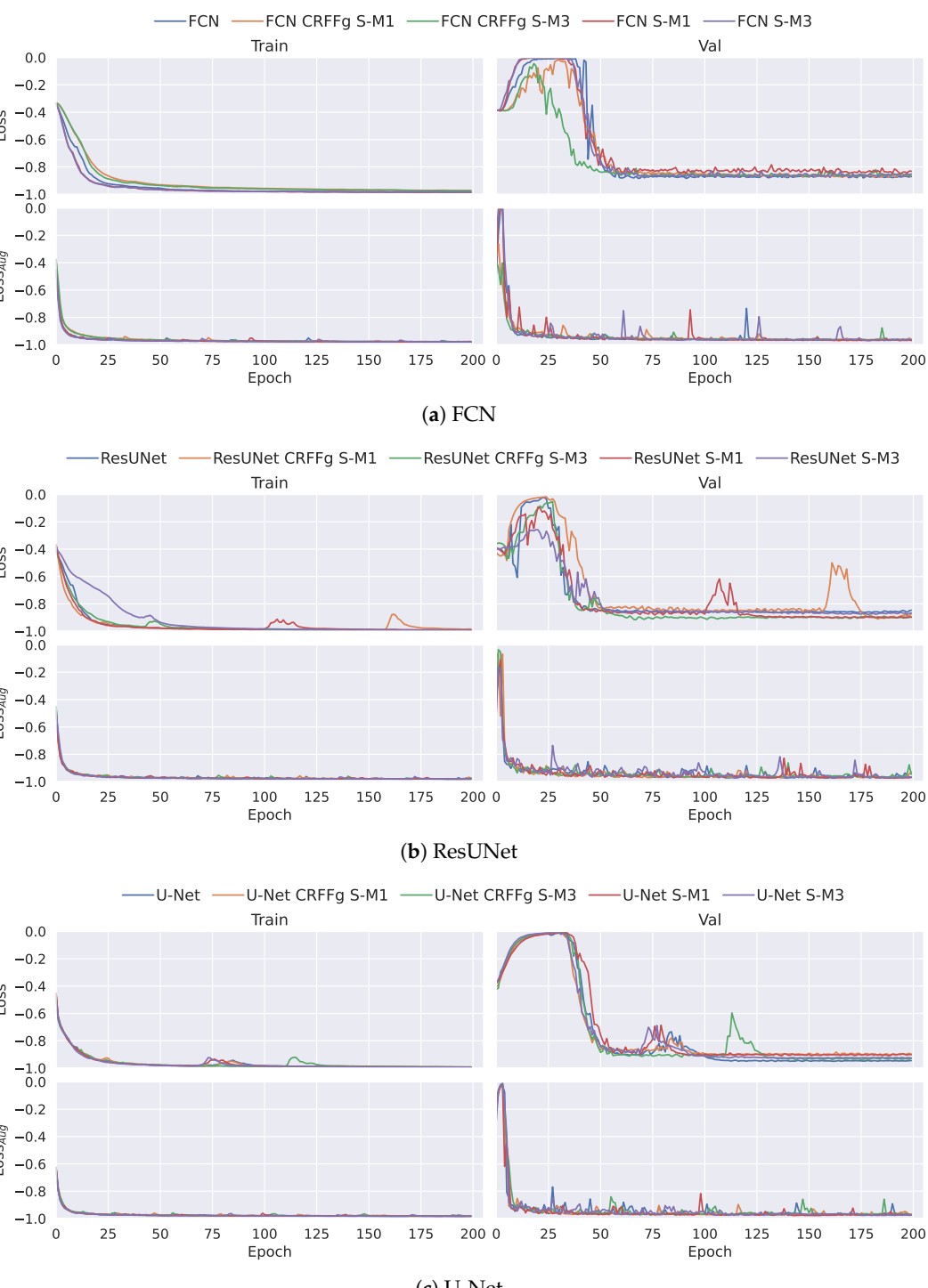

**Figure 9.** Training neural network loss vs. epochs corresponding to the various models examined are presented. M1 and M3 signify the dimensions of the CRFFg layer, expressed as multiplication factors of the enhanced layer's size.

It is worth noting that both the FCN CRFFg S-M3 and, to a lesser extent, the FCN CRFFg S-M1 tend to exhibit faster decreases in validation loss during early iterations. This conduct can be attributed to the generalization capabilities of the RFF from kernel methods. On the other hand, in the ResUNet architectures, although it needs to be clarified, the ResUNet S-M3 tends to experience an early decline, even though it also reaches its minimum early, which is not the minimum among the approaches. Conversely, no apparent differences are observed within the U-Net architectures. Notably, the models in the data augmentation scenario are similar.

In turn, Figure 10a displays the values of semantic segmentation performance for thermalFeet dataset achieved by each compared deep learning architecture: FCN (colored in blue), ResNet (red), U-Net (green). For interpretation purposes, the results are presented for the evaluation measures separately. As seen, the specificity estimates are very close to the maximal value and show the lowest variability. This result can be explained by the relatively small feet sizes compared with the background, making their correct detection and segmentation more difficult. On the contrary, sensitivity assessments are of less value and have much more variability, accounting for the diversity in the regions of interest (i.e., size, shape, and location). Due to the changing behavior of thermal patterns and the limited datasets available, learners have difficulty obtaining an accurate model.

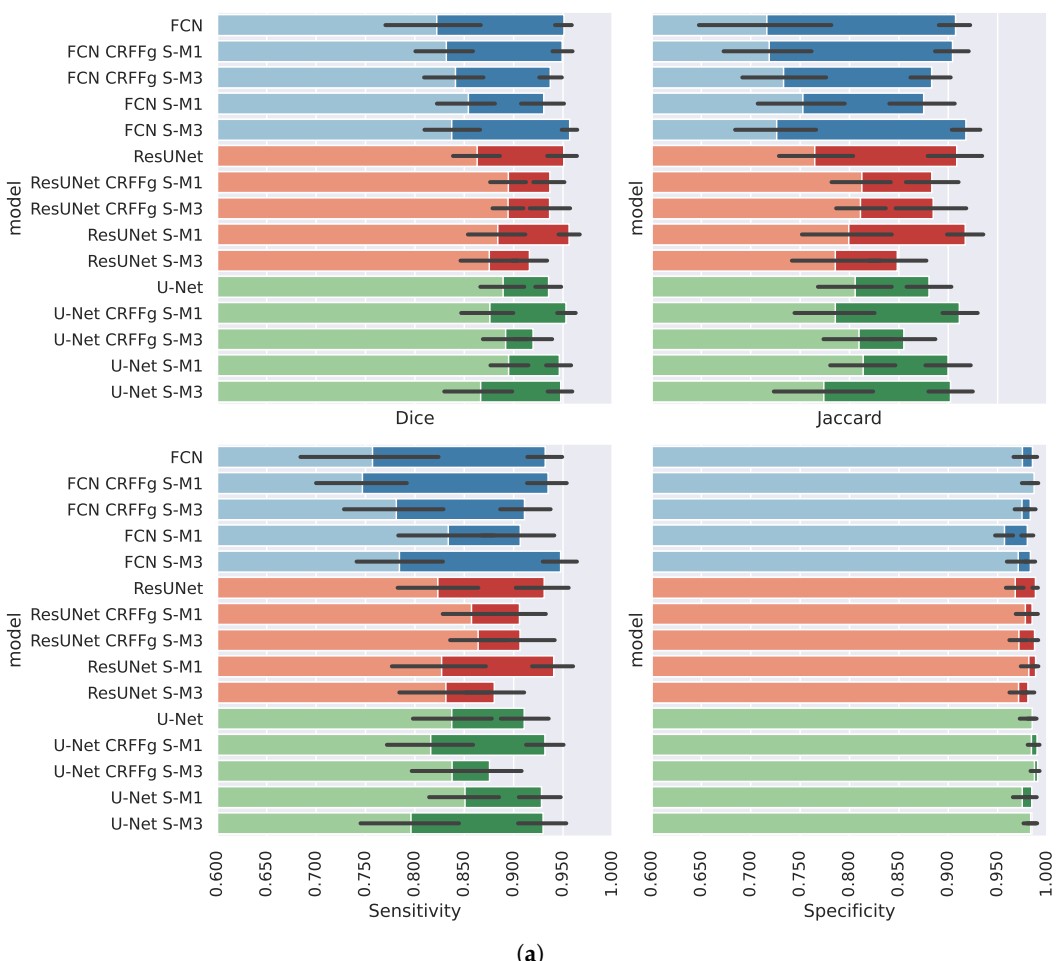

**(a)**

**Figure 10.** *Cont.*

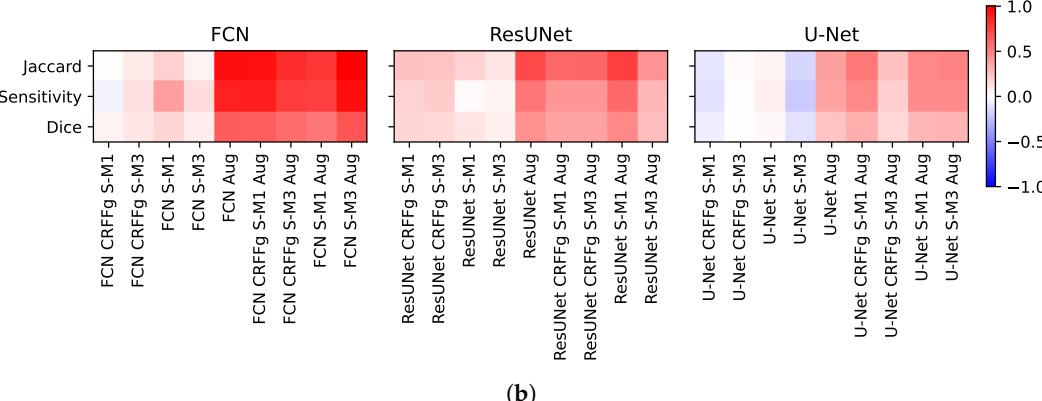

(**b**)

**Figure 10.** Results of the comparison between methods. The segmentation performance of Ther-malFeet is evaluated using baseline models FCN, UNet, and ResUNet, and compared to our proposal that incorporates CRFFg-based enhancements. M1 and M3 represent CRFFg's dimension as a multi-plication factor of the enhanced layer's size. Aug stands for data augmentation. (**a**) Segmentation performance results on ThermalFeet database. The three types of architecture used in this study (FCN, U-Net, ResUNet) are differentiated by color. The type of variation in the architecture is indicated by the marker used; (**b**) The improvement of each strategy, normalized with respect to the baseline performance of each architecture.

Regarding overlapping between estimated thermal masks, the Dice value is acceptable but with higher variance values for FCN, implying that other tested models segment complex shapes more accurately. As expected, the Jaccard index mean values resemble the Dice assessments, although with increased variance, which highlights the mismatch between the ground truth and the predicted mask even more.

A comparison between the segmentation metric value achieved by the baseline archi-tecture (without any modifications) and the value estimated for every evaluated semantic segmentation strategy is presented in Figure 10b. Note that specificity is removed because its estimates are obtained with minimal variations.

As seen, the performance improvement depends on the learner model size (also called algorithm complexity). Namely, the baseline architecture of FCN holds 1,197,375 param-eters, baseline ResUnet— 643,549, and baseline Unet—494,093. Thus, the FCN model contains the largest tuning parameter set and achieves the poorest performance, but it benefits the most from the evaluated architectures. As data augmentation is also applied, this finding becomes more evident. It may be pointed out that adding new data decreases model overfitting inherent to massive model sizes. Likewise, the following ResUnet model takes advantage of the enhanced architecture strategy using our CRFFg and improves performance. It increases more by generating new data points, however, to a lesser extent. Lastly, the learner with the lowest parameter set gets almost no benefits or is negatively affected by the strategies considered for architecture enhancement. Still, the strategies taken into account combined with expanded training data sizes can be improved, though very modestly. See Table A1, Appendix A, for the detailed segmentation performance results concerning the studied approaches.

### 4.3. Results of Assessing the Proposed CAM-Based Relevance Analysis Measures

We aim to evaluate the tested deep learning models for assessing the contribution of CAM-based representations to interpretability. To this end, we plot the pairwise rela-tionship between the essential explanation elements (background and foreground) and the above-proposed measure for assessing the CAM-based relevance of performed image segmentation masks. Figure 11 displays the scatter plots obtained by each segmentation learner. CAMs extracted by the learner contribute more to the interpretability of regions of interest if the measure value tends toward the top-right corner. Moreover, we focus on the

contribution of CAM representations to segmenting between background and foreground, utilizing the patient's feet as critical identification features.

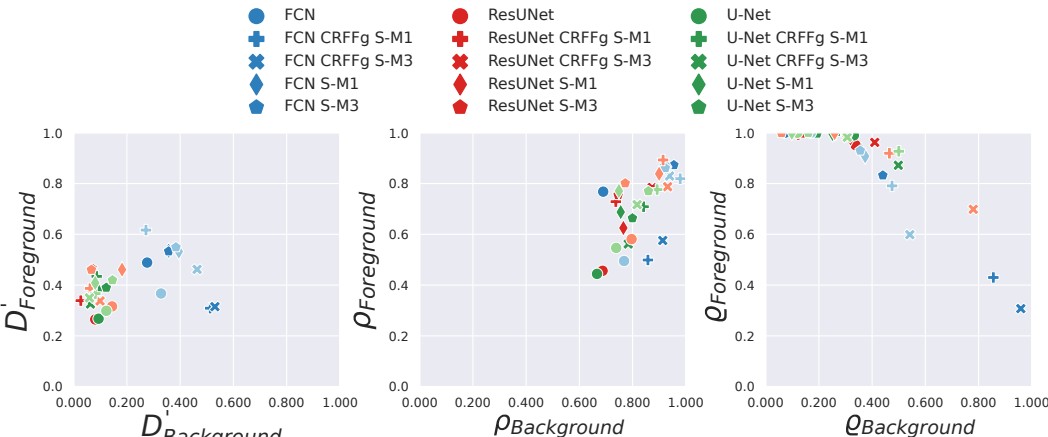

**Figure 11.** Results of Interpretability Measures on ThermalFeet. The three types of architecture used in this study (FCN, U-Net, ResUNet) are differentiated by color. The type of variation in the architecture is indicated by the marker used. M1 and M3 represent CRFFg's dimension as a multiplication factor of the enhanced layer's size.

The findings from the modified CAM-Dice results can be split into two groups (refer to the left plot in Figure 11). One group involves ResUnet and UNet architectures, and the other showcases the better performance, featuring FCN architectures. It is also important to mention that the data augmentation strategy does not significantly boost interpretability as much as it enhances segmentation performance measures. Looking at the CAM-based Cumulative Relevance (refer to the middle plot in Figure 11), it is apparent that models with refined representations at skip connections surpass the baseline models. Even though there is no substantial difference between models with these enhancements, most models are situated in the top-right corner. This position suggests that the primary relevance is focused on the area of interest. Significantly, relevance seems to accumulate more in the background than in the foreground, which is logical, considering the relative sizes of both areas. In Figure 11, the Mask-based Cumulative Relevance plot on the right side demonstrates that most models tend to exhibit high-foreground-low-background relevance. This pattern leads to a bias favoring the foreground class, as reflected in the more robust activation of CAMs for the foreground class. However, it is interesting that models employing CRFFg perform better in separating classes situated towards the top-right corner, suggesting superior capabilities in differentiating foreground and background classes.

Figure 12 displays examples of CAMs extracted by the best models per architecture under the Mask-based Cumulative Relevance for feet (colored in green) and background (red color), respectively. As seen, the higher weight is located at the last part of the decoder, where the higher values of semantic information are found. Besides, the weights for the background class are also less than for the foreground class, showing that the models emphasize the latter while preserving the relevance weights for the former.

In particular, FCN CRFFg S-M3 is the best FCN model, as shown in Figure 12a, and extracts most of the weights in three layers (i.e., l3, l4, and l5), meaning that other layers do not contribute to the class foreground. On the other hand, this architecture leads to CAMs with lower values for background class (see examples on the right). This behavior can be explained because the FCN architecture holds an extensive receptive field. Hence, the FCN CRFFg S-M3 model enables capturing more global information crucial for segmentation and concentrating weights in a few layers.

In the case of ResUNet, ResUNet CRFFg S-M3 performs the most efficiently, as shown in Figure 12b. Since the receptive field decreases, the ResUNet architecture distributes the contribution more evenly among the extracted CAM representations. However, the more

significant values remain in the l3, l4, and l5 layers. There is also activation of weights for the background class that can be explained, firstly, since the CRFFg configuration helps capture complex non-linear dependencies. Secondly, the local receptive field allows class separation.

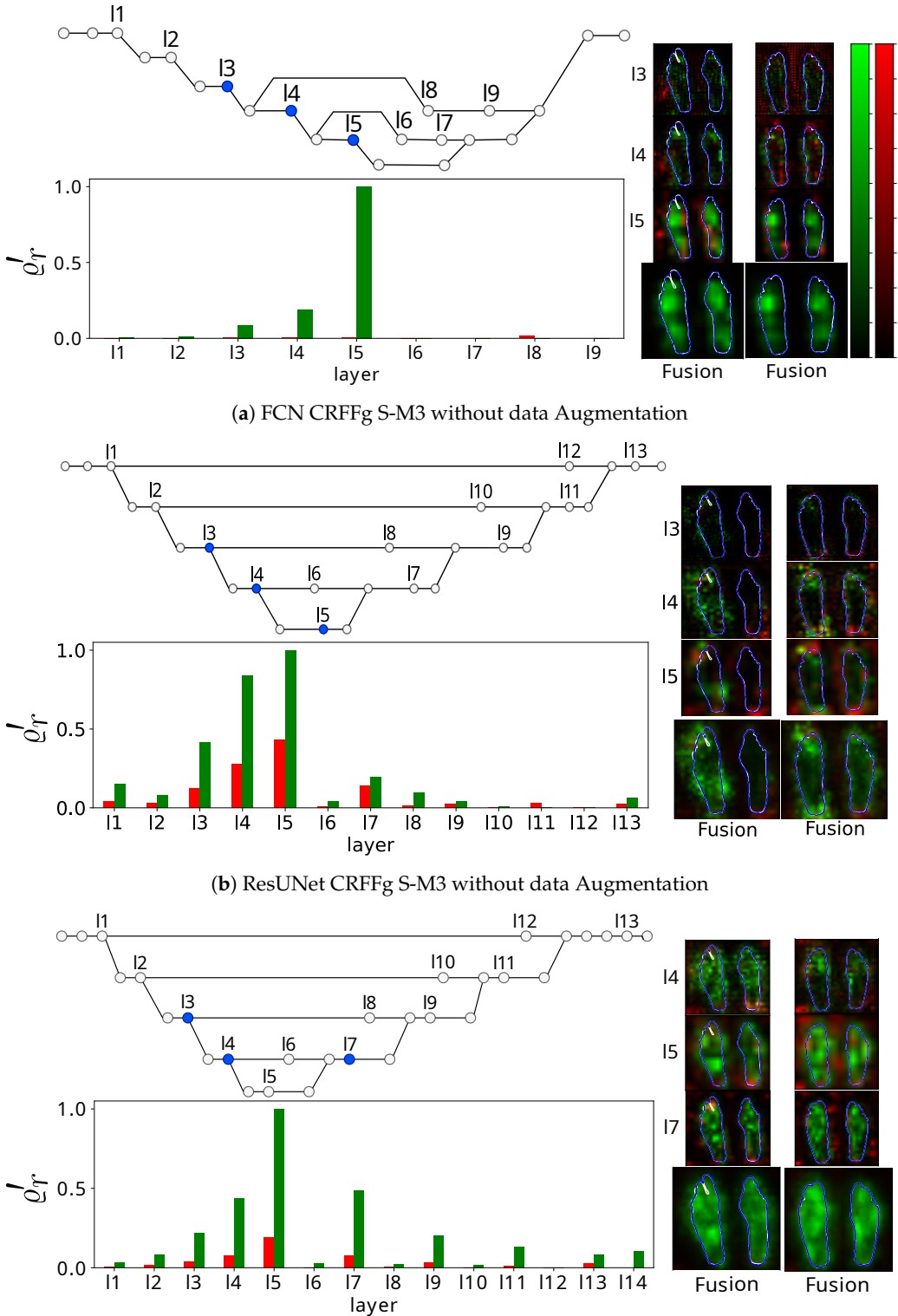

(**a**) FCN CRFFg S-M3 without data Augmentation

(**b**) ResUNet CRFFg S-M3 without data Augmentation

(**c**) U-Net CRFFg S-M3 with data Augmentation

**Figure 12.** Salient relevance analysis results. Best models concerning the Mask-based Cumulative Relevance, $\varrho_r$ measure, are presented for FCN, UNet, and ResUNet with our CRFFg-based enhancement.

Lastly, the CRFFg S-M3 model is the most effective for the U-Net architecture, with a performance similar to the outperforming ResUNet architecture, as shown in Figure 12c. However, several differences in the Fusion CAMs extracted by U-Net CRFFg S-M3 show high activation within the feet, suggesting that this model is not only sensitive to the foreground class. In addition, it captures more global features from feet.

## 5. Concluding Remarks

We introduce an innovative semantic segmentation approach that enhances interpretability by incorporating Convolutional Random Fourier Features and layer-wise weighted class activation maps. Our approach has been tested on a unique dataset of thermal foot images from pregnant women who have received epidural anesthesia, which is small but exhibits considerable variability. Besides, our strategy is two-pronged. Firstly, we introduce a novel Random Fourier Features layer, CRFFg, for handling image data, aiming to enhance three renowned architectures - FCN, UNet, and ResUNet. Secondly, we introduce three new quantitative measures to assess the interpretability of any deep learning model used for segmentation tasks. Our validation results indicate that the proposed approach boosts explainability and maintains competitive foot segmentation performance. In addition, the dataset used is tailored explicitly for epidural insertion during childbirth, reinforcing the practical relevance of our methodology.

There are, however, several observations worth mentioning:

Data acquisition tailored for Epidural. Epidural anesthesia involves the delivery of medicines that numb body parts to relieve pain, and the acquisition of data is usually performed under uncontrolled conditions with strong maternal artifacts. Moreover, it is impossible to fix a timeline for data collection. In addition, a timeline for gathering data cannot be set correctly. To the extent of our knowledge, this is the first time a protocol has been presented to regulate the data collection of infrared thermal images acquired from pregnant women who underwent epidural anesthesia during labor. As a result, data were assembled under real-world conditions that contained 196 thermal images fulfilling validation quality criteria.

Deep learning models for image semantic segmentation. Combined with machine learning, thermal imaging has proven helpful for performing semantic segmentation as a powerful method of dense prediction to adverse lighting conditions, providing better performance compared to their traditional counterparts. State-of-the-art medical image segmentation models include variants of U-Net models. A major reason for their success is that they employ skip connections, combining deep, semantic, and coarse-grained feature maps from the decoder subnetwork with shallow, low-level, fine-grained feature maps from the encoder subnetwork. They recover fine-grained details of target objects despite complex backgrounds [57]. Nevertheless, the collected image data from epidural anesthesia is insufficient for training the most commonly-known deep learners, which may result in overfitness to the training set. We address this issue by employing data augmentation addresses that artificially increase training data inputs to feed three tested architectures of deep learning models (FCN, U-Net, ResUNet), thus improving segmentation accuracy results. As seen in Figure 10b, the segmentation accuracy gain depends on the learner model complexity used: The fewer parameters the learner holds, the more the effectivity of data augmentation. Thus, the UNet learner with the lowest parameter set gets almost no benefit.

Strategies for enhancing the performance of deep learning-based segmentation. Three deep-learning architectures are explored to increase the interpretability of semantic segmentation results at competitive accuracy, ranked in decreased order of computational complexity as follows: FCN, ResUNet, and U-Net. Regarding the accuracy of semantic models, the data augmentation yields a sensibility metric value dependent on the model complexity: the more parameters the architecture holds, the higher the segmentation accuracy improvement. Thus, FCN benefits more from artificial data than ResUNet and U-Net. In the same way, both overlapping metrics (Jaccard and Dice) depend on the complexity of models. By contrast, the specificity reaches very high values regardless of trained deep

learning because the background texture's homogeneity saturates most captured thermal images. Nonetheless, the proposed modifications to architectures are not a solid argument for influencing their performed accuracy of semantic segmentation. In terms of enhancing explainability, the weak influence of data augmentation is the first finding to be drawn, as seen in the scatterplots of Figure 11. All tested models produce more significant CAM activations from layers with a wider receptive field. Moreover, the CRFFg layer also improves the representation of the foreground and background. It is also important to note the metrics developed for assessing the explainability of CAM representations, allowing scalability to larger image sets without visual inspection.

In terms of future research, the authors intend to integrate Vision Transformers and attention mechanisms for semantic segmentation into the CRFFg-based representation [58]. Besides, we propose to include variational autoencoders and transfer learning strategies within our framework to prevent overfitting and enhance data interpretability [33,59].

**Author Contributions:** Conceptualization, J.C.A.-A., A.M.Á.-M. and G.C.-D.; methodology, J.C.A.-A., A.M.Á.-M. and G.C.-D.; software, J.C.A.-A.; validation, J.C.A.-A., A.M.Á.-M. and G.C.-D.; formal analysis, J.C.A.-A. and G.C.-D.; investigation, J.C.A.-A., A.M.Á.-M. and G.C.-D.; resources, A.M.Á.-M. and G.C.-D.; data curation, J.C.A.-A.; writing–original draft preparation, J.C.A.-A., A.M.Á.-M. and G.C.-D.; writing–review and editing, A.M.Á.-M. and G.C.-D.; visualization, J.C.A.-A.; supervision, A.M.Á.-M. and G.C.-D.; project administration, A.M.Á.-M.; funding acquisition, A.M.Á.-M. and G.C.-D. All authors have read and agreed to the published version of the manuscript.

**Funding:** Under grants provided by the projects: "Prototipo de visión por computador para la identificación de problemas fitosanitarios en cultivos de plátano en el departamento de Caldas" (Hermes 51175) funded by Universidad Nacional de Colombia, and "Desarrollo de una herramienta de visión por computador para el análisis de plantas orientado al fortalecimiento de la seguridad alimentaria" (Hermes 54339) funded by Universidad Nacional de Colombia and Universidad de Caldas.

**Institutional Review Board Statement:** This study uses anonymized public datasets with institutional review board statement as presented in https://gcpds-image-segmentation.readthedocs.io/en/latest/notebooks/02-datasets.html (accessed on 5 April 2023).

**Informed Consent Statement:** This study uses anonymized public datasets as presented in https://gcpds-image-segmentation.readthedocs.io/en/latest/notebooks/02-datasets.html (accessed on 5 April 2023).

**Data Availability Statement:** Dataset is publicly available at: https://gcpds-image-segmentation.readthedocs.io/en/latest/notebooks/02-datasets.html (accessed on 5 April 2023).

**Conflicts of Interest:** The authors declare no conflict of interest.

## Appendix A. Method Comparison from Absolute Semantic Segmentation Performances

Table A1 presents the absolute semantic segmentation results acquired from thermalFeet database. For clarity, the rank position of each method is also included. As can be seen, our enhancement based on CRFFg boosts the segmentation performance. Notably, ResUNet CRFFg S-M1 outperforms the tested approaches concerning the measured quantitative assessments.

**Table A1.** Absolute Semantic Segmentation Results on thermalFeet database. WODA: Without Data Augmentation, WDA: With Data Augmentation. The average ± standard deviation performance is displayed regarding the test partitions. M1 and M3 stand for CRFFg's dimension as a multiplication factor of the enhanced layer's size.

| Approach | Measure | WODA | Rank | WDA | Rank |
|---|---|---|---|---|---|
| FCN | Dice | 0.9527 ± 0.0238 | 3.0 | 0.8646 ± 0.0624 | 10.0 |
| | Jaccard | 0.9106 ± 0.0424 | 3.0 | 0.7668 ± 0.0969 | 10.0 |
| | Sensitivity | 0.9352 ± 0.0482 | 4.0 | 0.8260 ± 0.1098 | 6.0 |
| | Specificity | 0.9857 ± 0.0105 | 7.0 | 0.9697 ± 0.0186 | 13.0 |
| FCN CRFFg S-M1 | Dice | 0.9530 ± 0.0257 | 2.0 | 0.8510 ± 0.0623 | 12.0 |
| | Jaccard | 0.9113 ± 0.0456 | 2.0 | 0.7456 ± 0.0913 | 12.0 |
| | Sensitivity | 0.9424 ± 0.0526 | 3.0 | 0.8016 ± 0.0999 | 13.0 |
| | Specificity | 0.9810 ± 0.0158 | 12.0 | 0.9697 ± 0.0233 | 14.0 |
| FCN CRFFg S-M3 | Dice | 0.9480 ± 0.0224 | 5.0 | 0.8346 ± 0.0916 | 15.0 |
| | Jaccard | 0.9021 ± 0.0403 | 5.0 | 0.7262 ± 0.1284 | 15.0 |
| | Sensitivity | 0.9340 ± 0.0423 | 6.0 | 0.7771 ± 0.1325 | 15.0 |
| | Specificity | 0.9804 ± 0.0168 | 13.0 | 0.9714 ± 0.0246 | 10.0 |
| FCN S-M1 | Dice | 0.9469 ± 0.0273 | 6.0 | 0.8421 ± 0.0870 | 14.0 |
| | Jaccard | 0.9003 ± 0.0486 | 6.0 | 0.7367 ± 0.1254 | 14.0 |
| | Sensitivity | 0.9286 ± 0.0518 | 7.0 | 0.7867 ± 0.1422 | 14.0 |
| | Specificity | 0.9843 ± 0.0109 | 9.0 | 0.9714 ± 0.0207 | 9.0 |
| FCN S-M3 | Dice | 0.9519 ± 0.0281 | 4.0 | 0.8470 ± 0.0737 | 13.0 |
| | Jaccard | 0.9096 ± 0.0499 | 4.0 | 0.7414 ± 0.1070 | 13.0 |
| | Sensitivity | 0.9341 ± 0.0543 | 5.0 | 0.8160 ± 0.1152 | 9.0 |
| | Specificity | 0.9865 ± 0.0107 | 6.0 | 0.9604 ± 0.0300 | 15.0 |
| ResUNet | Dice | 0.9348 ± 0.0502 | 11.0 | 0.8569 ± 0.0779 | 11.0 |
| | Jaccard | 0.8816 ± 0.0868 | 11.0 | 0.7575 ± 0.1152 | 11.0 |
| | Sensitivity | 0.9029 ± 0.0825 | 12.0 | 0.8152 ± 0.1316 | 11.0 |
| | Specificity | 0.9896 ± 0.0067 | 2.0 | 0.9712 ± 0.0180 | 12.0 |
| ResUNet CRFFg S-M1 | Dice | 0.9456 ± 0.0317 | 7.0 | 0.8851 ± 0.0449 | 4.0 |
| | Jaccard | 0.8984 ± 0.0560 | 7.0 | 0.7968 ± 0.0709 | 4.0 |
| | Sensitivity | 0.9472 ± 0.0540 | 1.0 | 0.8283 ± 0.0853 | 5.0 |
| | Specificity | 0.9725 ± 0.0230 | 14.0 | 0.9841 ± 0.0123 | 3.0 |
| ResUNet CRFFg S-M3 | Dice | 0.9111 ± 0.0602 | 15.0 | 0.8969 ± 0.0444 | **1.0** |
| | Jaccard | 0.8420 ± 0.0951 | 15.0 | 0.8160 ± 0.0737 | 1.0 |
| | Sensitivity | 0.9075 ± 0.0607 | 11.0 | 0.8675 ± 0.0803 | 1.0 |
| | Specificity | 0.9663 ± 0.0346 | 15.0 | 0.9712 ± 0.0244 | 11.0 |
| ResUNet S-M1 | Dice | 0.9558 ± 0.0279 | 1.0 | 0.8865 ± 0.0676 | 3.0 |
| | Jaccard | 0.9167 ± 0.0498 | 1.0 | 0.8026 ± 0.1061 | 3.0 |
| | Sensitivity | 0.9459 ± 0.0482 | 2.0 | 0.8403 ± 0.1123 | 2.0 |
| | Specificity | 0.9831 ± 0.0152 | 10.0 | 0.9750 ± 0.0287 | 8.0 |
| ResUNet S-M3 | Dice | 0.9237 ± 0.0411 | 14.0 | 0.8677 ± 0.0894 | 9.0 |
| | Jaccard | 0.8610 ± 0.0713 | 14.0 | 0.7763 ± 0.1281 | 9.0 |
| | Sensitivity | 0.8875 ± 0.0756 | 14.0 | 0.8179 ± 0.1333 | 8.0 |
| | Specificity | 0.9846 ± 0.0128 | 8.0 | 0.9755 ± 0.0217 | 7.0 |
| U-Net | Dice | 0.9371 ± 0.0312 | 10.0 | 0.8713 ± 0.0756 | 8.0 |
| | Jaccard | 0.8832 ± 0.0551 | 10.0 | 0.7796 ± 0.1145 | 8.0 |
| | Sensitivity | 0.9120 ± 0.0571 | 10.0 | 0.8107 ± 0.1248 | 12.0 |
| | Specificity | 0.9811 ± 0.0199 | 11.0 | 0.9847 ± 0.0130 | 2.0 |
| U-Net CRFFg S-M1 | Dice | 0.9448 ± 0.0297 | 8.0 | 0.8827 ± 0.0617 | 5.0 |
| | Jaccard | 0.8969 ± 0.0528 | 8.0 | 0.7954 ± 0.0965 | 5.0 |
| | Sensitivity | 0.9160 ± 0.0561 | 9.0 | 0.8383 ± 0.1062 | 4.0 |
| | Specificity | 0.9902 ± 0.0057 | 1.0 | 0.9780 ± 0.0124 | 5.0 |

**Table A1.** *Cont.*

| Approach | Measure | WODA | Rank | WDA | Rank |
|---|---|---|---|---|---|
| U-Net CRFFg S-M3 | Dice | 0.9252 ± 0.0404 | 13.0 | 0.8821 ± 0.0645 | 6.0 |
| | Jaccard | 0.8634 ± 0.0694 | 13.0 | 0.7948 ± 0.1004 | 6.0 |
| | Sensitivity | 0.8831 ± 0.0730 | 15.0 | 0.8231 ± 0.1110 | 7.0 |
| | Specificity | 0.9893 ± 0.0066 | 3.0 | 0.9873 ± 0.0088 | 1.0 |
| U-Net S-M1 | Dice | 0.9400 ± 0.0364 | 9.0 | 0.8898 ± 0.0536 | 2.0 |
| | Jaccard | 0.8890 ± 0.0635 | 9.0 | 0.8056 ± 0.0861 | 2.0 |
| | Sensitivity | 0.9162 ± 0.0619 | 8.0 | 0.8384 ± 0.0904 | 3.0 |
| | Specificity | 0.9866 ± 0.0086 | 5.0 | 0.9777 ± 0.0208 | 6.0 |
| U-Net S-M3 | Dice | 0.9293 ± 0.0419 | 12.0 | 0.8767 ± 0.0772 | 7.0 |
| | Jaccard | 0.8707 ± 0.0728 | 12.0 | 0.7883 ± 0.1152 | 7.0 |
| | Sensitivity | 0.8934 ± 0.0792 | 13.0 | 0.8152 ± 0.1181 | 10.0 |
| | Specificity | 0.9878 ± 0.0098 | 4.0 | 0.9805 ± 0.0189 | 4.0 |

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
