# Peer review of "Feet Segmentation for Regional Analgesia Monitoring Using Convolutional RFF and Layer-Wise Weighted CAM Interpretability"

_computation, doi:10.3390/computation11060113_

Round 1

Reviewer 1 Report

The authors has shown good effort on developed a new approach for the issue. This paper proposes the CRFFg technique to improve segmentation performance based on three image segmentation-specific convolutional models. Here are some comments:

The paper discusses visual transformers but only rejects evaluating any ViT-based models. It would be best if this manuscript could evaluate some ViT-based models and show that ViT underperforms under your hypothesis.

This manuscript paper should include showing a figure of an example of the final segmentation results of different models to demonstrate the segmentation performance of all models compared to the ground truth.

Network fine-tuning strategy, hyperparameters used in each network, and computer equipment exploited should be provided.

Some of the figures do not provide units, while Figure 6 is too compact.

Figure 4(b) only shows the relative performance improvement corresponding to the base model. How about the absolute Dice and Jaccard performance comparison?

The quality is fine.

Reviewer 2 Report

The contribution of this work is not so much clear. I suggest to highlight the contributions concisely in abstract, and mention those in introduction. Moreover, reflect the result analysis based on your contribution. Please revise the conclusion also. 

Reviewer 3 Report

In the present paper an innovative semantic image segmentation methodology using Convolutional Random Fourier Features for foot segmentation is proposed. In the method, a random fourier features layer is introduced to deal with images to enhance the FCN, UNet and ResUNet architectures. In general, the paper is well written in English and it presents an informative state of the art, which is pertinent to introduce the problem

My main concerns with the present research are the following:

a) Please check some typos such as: quantitive (line 17),CRRg (line 255), segmantic, architectuhres (caption of Figure 3), Morevoer, [?] (line 290), meeasure (line 337)
b) Please increase the number of comparative methods for foot segmentation.
c) Please explain if the transfer learning strategy can be used to improve the comparative analysis.
d) Please use more deep learning techniques for test the proposed CRFFg strategy.
e) Explain and present how the parameters of the different methods were tuned.
f) Please explain the selection of the Adam optimizer.
g) Please present the plot of convergence (efficiency vs iterations) of the improved methods.
h) Please improve the quality of Figure 5.

In general, the paper is well written in English, but please check some typos such as:

quantitive (line 17),

CRRg (line 255),

segmantic, architectuhres (caption of Figure 3),

Morevoer, [?] (line 290), meeasure (line 337)

Reviewer 4 Report

The authors proposed a feet segmentation method regional analgesia monitoring. Novel Random Fourier Features layers are introduced to three well-known architectures (FCN, UNet, and ResUNet) to improve the performance. Three novel quantitative measures are proposed to evaluate the methods for this specific task. Tests on a public data set demonstrated the effectiveness of the proposed method.

Minor comments:

1. Better to move Figure 1 and the general introduction of the whole pipeline to chapter 2.

2. Reference error in Line 290.

3. Please keep the naming of the methods consistent through the manuscript, e.g., UNet/Unet/U-Net, ResUnet/ResUNet

4. Line 100: should Convolutional Random Fourier Features (CRFFg) be written as CRFF or Convolutional Random Fourier Features Gradient?

5. Equ(13), what does the denominator ρc  represent?

6. How did the authors prepared the ground truth? Were they manually labelled?

7. Line 17: quantitive -> quantitative

Round 2

Reviewer 2 Report

My concerns are addressed. 

Reviewer 3 Report

My comments have been properly addressed.

The paper is well written in English.